# Queuosine is incorporated into precursor tRNA before splicing

Wei Guo [1,2,3,11], Igor Kaczmarczyk [4,5,11], Kevin Kopietz [6], Florian Flegler[7], Stefano Russo [1,2,3], Ege Cigirgan[8], Andrzej Chramiec-Głąbik[4], Łukasz Koziej [4], Cansu Cirzi[9], Jirka Peschek [8], Klaus Reuter[7], Mark Helm [6], Sebastian Glatt [4,10] & Francesca Tuorto [2,3] ✉

Each newly transcribed tRNA molecule must undergo processing and receive modifications to become functional. Queuosine (Q) is a tRNA modification present at position 34 of four tRNAs with "GUN" anticodons. Among these, the precursor of tRNA[Tyr] carries an intronic sequence within the anticodon loop that is removed by an essential non-canonical splicing event. The functional and temporal coupling between tRNA-splicing and Q-incorporation remains elusive. Here, we demonstrate in vitro and in vivo that intron-containing precursors of tRNA[Tyr] are modified with Q or with the Q-derivative galactosyl-queuosine (galQ) before being spliced. We show that this order of events is conserved in mouse, human, flies and worms. Using single particle cryo-EM, we confirm that pre-tRNA[Tyr] is a bona fide substrate of the QTRT1/2 complex, which catalyzes the incorporation of Q into the tRNA. Our results elucidate the hierarchical interplay that coordinates Q-incorporation and splicing in eukaryotic tRNAs, providing a relevant but unappreciated aspect of the cellular tRNA maturation process.

Transfer RNAs (tRNAs) function as adaptor molecules during protein synthesis, translating the information encoded in mRNAs into nascent polypeptide chains. Beyond protein synthesis, tRNAs are also involved in several other cellular processes, ranging from stress response, metabolic sensing, and viral infections to non-Mendelian genetic inheritance[1-4]. The capacity and versatility of these adaptors are greatly influenced by post-transcriptional processing and modifications[5]. Among all RNA molecules, tRNA is the most diverse and heavily modified one with more than 90 different types of modifications so far detected in tRNAs of all domains of life[6]. The incorporation of specific nucleotide modifications at discrete positions is key to tRNA function.

To become functionally mature tRNAs, each tRNA transcript needs to go through several maturation steps, comprising the removal of the 5′-leader and 3′-trailer sequences, the addition of the universal CCA-tail at the 3′ end, and for a subset of tRNAs, the removal (also known as splicing) of an intronic sequence stretch[5]. The precise hierarchical and temporal order as well as spatial organization of these processing steps is largely unknown.

In most bacteria and eukaryotes, queuosine (Q) is universally found at the wobble position (position 34) of four specific tRNAs featuring "GUN" anticodon sequences, namely tRNA[Tyr]$_{GUA}$, tRNA[Asp]$_{GUC}$, tRNA[Asn]$_{GUU}$, and tRNA[His]$_{GUG}$[7,8]. Prokaryotes can synthesize Q from GTP

[1]Faculty of Biosciences, Heidelberg University, Heidelberg, Germany. [2]Center for Molecular Biology of Heidelberg University (ZMBH), DKFZ-ZMBH Alliance, Heidelberg, Germany. [3]Division of Biochemistry, Mannheim Institute for Innate Immunoscience (MI3), Mannheim Cancer Center (MCC), Medical Faculty Mannheim, Heidelberg University, Mannheim, Germany. [4]Małopolska Centre of Biotechnology, Jagiellonian University, Kraków, Poland. [5]Doctoral School of Exact and Natural Sciences, Jagiellonian University, Kraków, Poland. [6]Institute of Pharmaceutical and Biomedical Science (IPBS), Johannes Gutenberg-University Mainz, Mainz, Germany. [7]Institut für Pharmazeutische Chemie, Philipps-Universität Marburg, Marburg, Germany. [8]Biochemistry Center (BZH), Heidelberg University, Heidelberg, Germany. [9]Division of Epigenetics, DKFZ-ZMBH Alliance, German Cancer Research Center (DKFZ), Heidelberg, Germany. [10]Department for Biological Sciences and Pathobiology, University of Veterinary Medicine Vienna, Vienna, Austria. [11]These authors contributed equally: Wei Guo, Igor Kaczmarczyk. ✉e-mail: francesca.tuorto@medma.uni-heidelberg.de

de novo, whereas eukaryotes need to salvage the modified nucleoside or the base queuine (q) from their diet and microbiota[9]. In eukaryotes, the incorporation of q into tRNAs is carried out by the tRNA guanine transglycosylase complex (TGT), comprising the catalytic queuine tRNA-ribosyltransferase subunit 1 (QTRT1), and the accessory subunit QTRT2[10]. The two subunits form a heterodimer (QTRT1/2), which binds and incorporates q at position 34 of the target tRNAs[11,12]. For tRNA$^{Tyr}$ and tRNA$^{Asp}$, Q$_{34}$ is further modified by the addition of either a galactosyl or a mannosyl group by a galactosyltransferase (QTGAL) and a mannosyltransferase (QTMAN), respectively[13]. TGTs are present in all domains of life, with homologous structural traits and conserved catalytic mechanisms. However, archaeal TGTs modify position 15 in the D-loop of several tRNAs with archeosine instead of Q[14]. The Q modification at the wobble position increases the efficiency of decoding during translation[13,15]. The lack of Q has been linked to several neurological and psychiatric diseases, such as Parkinson's and Alzheimer's diseases, multiple sclerosis, and schizophrenia[16–18]. Notably, a mouse model lacking *Qtrt1* (Q1) revealed the role of Q in protein synthesis and the related defects in the neuronal cytoarchitecture and cognitive performance[15]. Q has been reported to be absent in intron-containing pre-tRNA$^{Tyr}$ in *Trypanosoma brucei* (*T. brucei*)[19], but the hierarchical interplay between splicing and Q modification has not yet been studied in other eukaryotes, including mouse and human.

In eukaryotes, intron-containing pre-tRNAs undergo non-conventional splicing[20]. The various isoforms of these pre-tRNAs are encoded at different gene loci and are transcribed with diverse intronic sequences, but result in an identical mature tRNA sequence. The removal of the intron, invariably located between positions 37 and 38 of the mature tRNA, occurs in a two-step process. First, the intron is excised by the splicing endonuclease complex called TSEN[21]. Second, the two resulting exon halves of the cleaved tRNA are sealed by the tRNA ligase RTCB with the assistance of Archease, DDX1, ASW, FAM98B and the contribution of CLP1 kinase to form the mature tRNA[22]. Mutations in different subunits of the TSEN complex and in CLP1 have been identified in patients carrying pontocerebellar hypoplasia, a heterogeneous group of neurodegenerative disorders with prenatal to neonatal onset characterized by cerebellar hypoplasia and microcephaly[23,24]. The accumulation of aberrant intermediate tRNA fragments derived from defective splicing has been suggested as the main cause of the pathological phenotypes[24,25]. The evolutionary conservation of both introns and site-specific modifications around the splice-site in individual tRNAs serves as an indicator of their evolved interconnection.

Here, we show that certain precursors of tRNA$^{Tyr}$ are modified with Q in various eukaryotic organisms, including *Mus musculus*, *Homo sapiens*, *Drosophila melanogaster* (*Dm*) and *Caenorhabditis elegans* (*C. elegans*), before splicing occurs. Structures of mouse QTRT1/2 in complex with mature tRNA$^{Tyr}$ and pre-tRNA$^{Tyr}$ by single particle cryo-Electron Microscopy (cryo-EM) confirm that the complex is able to bind both forms in an almost identical fashion. The structural analysis reveals that G$_{34}$ in pre-tRNA$^{Tyr}$ is positioned in the active site of QTRT1, whereas the intronic sequence remains mostly flexible. The presence of Q on pre-tRNA does not hamper the downstream splicing process, since endogenous pre-tRNA$^{Tyr}$ from wt and Q1 cells can be efficiently cut by TSEN. In summary, our results show that the intron-containing tRNA$^{Tyr}$ is a bona fide substrate of the QTRT1/2 complex in vitro and in vivo, suggesting that q-incorporation precedes tRNA splicing. Our work highlights the link between tRNA modification and processing, providing insight into the role of Q in health and disease.

## Results

### tRNA$^{Tyr}$ is queuosinylated at the precursor level in mESCs and brain tissues

A broad range of central and peripheral nervous system diseases are caused by mutations in genes encoding tRNA modification enzymes[26]

or components of the tRNA splicing machinery[27], pointing to a central function of tRNA splicing and protein translation in brain development and physiology. We sought to study the molecular interplay between tRNA splicing and Q modification by determining the levels of Q- and galQ-incorporation in tRNA$^{Tyr}$ at different stages of differentiation. For this purpose, mouse embryonic stem cells (mESCs), which can differentiate into various neuronal lineages were employed as a physiologically relevant model system. Based on its ability to react with cis-diol groups, acryloyl aminophenyl boronic acid (APB) combined with Northern blotting is a widely used approach to quantitatively determine the level of Q in discrete tRNA molecules[28,29] (Fig. 1a, b). In tRNA$^{Tyr}$, the 11-hydroxyl group of Q is further modified to galQ, so that its migration is no longer distinguishable from unmodified tRNAs in standard APB gels (Fig. 1a)[29,30]. Therefore, the modification of mature tRNA$^{Tyr}$ can easily be missed by this commonly used detection method[29–31]. Nevertheless, we could detect a small fraction (<4 %) of modified Q on mature tRNA$^{Tyr}$ in mESCs (Fig. 1c). In the mouse genome there are ten genes for tRNA$^{Tyr}_{GUA}$, which is the only anticodon decoding both UAC and UAU codons for Tyr. These isodecoders not only differ in their respective intronic sequences, but also in their expression levels (Fig. 1d, S1a, b). Using a pool of probes directed against the different intronic sequences, we detected the presence of Q in the pre-tRNAs before splicing as indicated by a shift in APB Northern blots (Fig. 1e). Our results showed that about 60% of pre-tRNA$^{Tyr}$ carries a Q modification. The specificity of the detection and the presence of Q in pre-tRNA is demonstrated by the absence of the signals in knockout cell lines, lacking *Qtrt1* (Q1) or *Qtrt2* (Q2), respectively. Furthermore, we performed APB Northern blots using probes against individual intronic sequences, confirming the presence of Q in specific pre-tRNAs. In detail, we detected Q in isoforms pre-tRNA$^{Tyr}$ 1-4 and pre-tRNA$^{Tyr}$ 2-1 in wild type mESCs (Fig. 1f) and pre-tRNA$^{Tyr}$1-4 in mouse brain tissue (Fig. 1g). However, the most abundantly expressed isoform, namely pre-tRNA$^{Tyr}$ 1-2, showed no APB-induced shift in mESCs or in mouse brain (Fig. 1g and Supplementary Fig. 1c).

Comparable levels of mature tRNA$^{Tyr}$ in wild type, Q1 and Q2 knockout cells exclude a strict Q-dependency of the splicing processes themselves. The results were confirmed in mESCs and brains from female or male mice (Supplementary Fig. 2). Interestingly, we observed an increase of pre-tRNA fragments in differentiated neurons lacking Q (Supplementary Fig. 2a). This finding is further corroborated by an increase of pre-tRNA$^{Tyr}$ 1-4 fragments in mouse tissues, including the brain (Supplementary Fig. 2c). Furthermore, unspliced pre-tRNA$^{Tyr}$ 1-4 can be localized in the cytoplasm (Supplementary Fig. 3a–c) suggesting that its presence outside the nucleus does not impair splicing. Consistently, we demonstrated that Q-incorporation does not affect the splicing process as native pre-tRNA$^{Tyr}$ 1-4 from wt and Q1 cells was efficiently cleaved by purified TSEN in vitro (Supplementary Fig. 3a, d, e).

Altogether, our findings reveal the unexpected substantial presence of Q-modified pre-tRNAs containing introns in vivo.

### Reconstitution of pre-tRNA$^{Tyr}$ queuosinylation in vitro

In order to confirm unspliced tRNA as target of the Q-modification, we purified the recombinant murine tRNA-guanine transglycosylase (mQTRT1/2) complex and reconstituted the modification reaction in vitro[11] (Supplementary Fig. 4). The enzymatic incorporation of q was measured using liquid chromatography-tandem mass spectrometry (LC-MS/MS) and APB Northern blot analysis with a variety of canonical and synthetic mutant oligoribonucleotides, corresponding to pre- and mature-tRNAs (Fig. 2a). In agreement with the data obtained in vivo, mouse pre-tRNA$^{Tyr}$ 1-4 and 2-1 were modified by purified mQTRT1/2. In contrast, mature and pre-tRNA$^{Leu}_{CAA}$ 2-1 showed no incorporation of q and therefore served as negative/specificity controls (Fig. 2a, b and Supplementary Fig. 4h). Enzymatic kinetics measurements ([8-$^3$H]-guanine incorporation assay (Supplementary Fig. 4a–g) and the analyses of covalent intermediate formation (Supplementary Fig. 4i) show

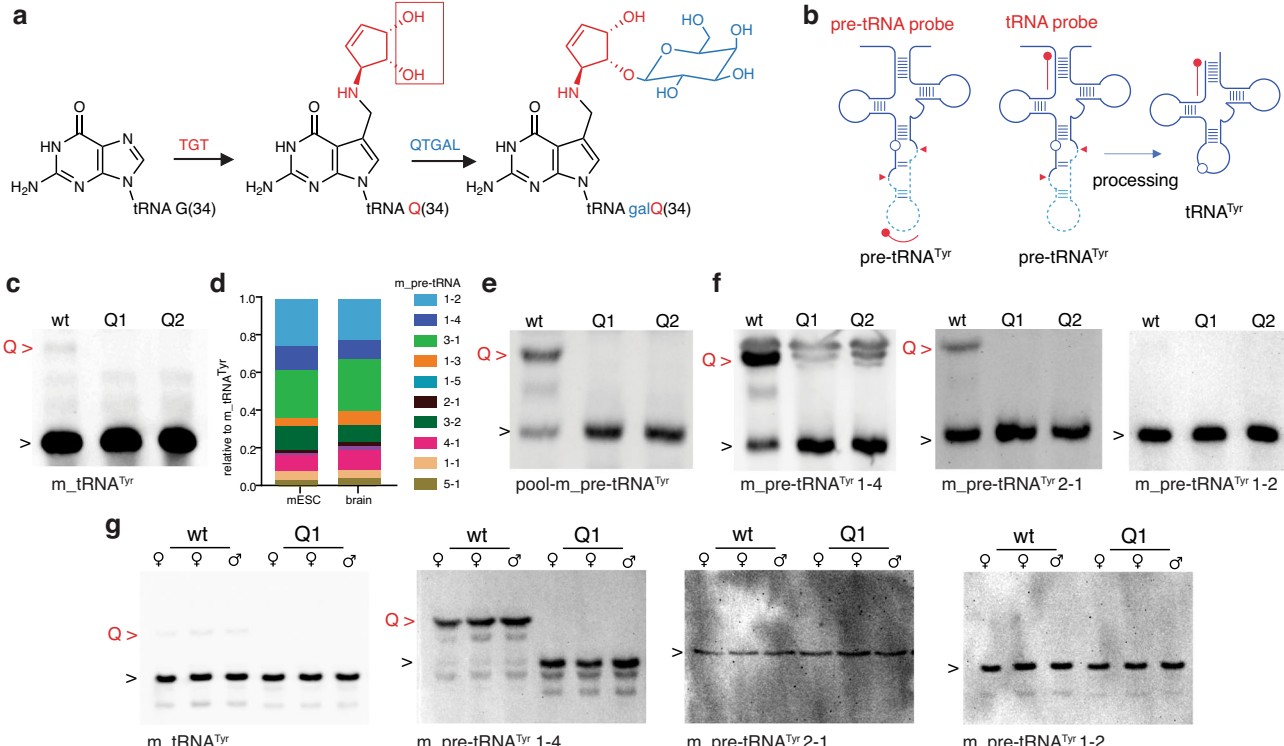

**Fig. 1 | tRNA^Tyr is queuosinylated at precursor level in mESCs and tissues. a** $G_{34}$ of tRNA^Tyr is modified with queuosine (Q) and galactosyl-queuosine (galQ) by the sequential action of the QTRT1/2 complex (TGT) and QTGAL enzyme. Red rectangle indicates the cis-diol group reacting in APB Northern blot. **b** Schematic representation of the probes used in the Northern blots mapped onto the standard cloverleaf structure of tRNAs. The empty circles represent $G_{34}$, the red triangles indicate the splicing sites. Pre-tRNA probes are designed complementary to the intron and detect only precursor tRNAs containing introns. tRNA probes are designed complementary to the 5′end of specific tRNAs and can detect both pre- and mature tRNAs. **c** GalQ on mature tRNA^Tyr cannot be resolved in APB Northern blot, which shows only residual Q modification in mESCs. **d** Expression of pre-tRNA^Tyr is quantified by Northern blot (Supplementary Fig. 1b) in mESCs and in the mouse brain. **e** APB Northern blot showing queuosinylation of tRNA^Tyr precursors in wild type mESCs with a pool of probes targeting pre-tRNA^Tyr 1-5, 2-1, 3-2. *Qtrt1*(Q1) and *Qtrt2* (Q2) knockout cell lines are shown as negative control. **f** Specific probes against mouse pre-tRNA^Tyr 1-4 (left panel) and pre-tRNA^Tyr 2-1 (middle panel) show that these specific precursors are modified already at precursor level in wild type mESC. **g** Specific probes against mouse tRNA^Tyr precursors show that pre-tRNA^Tyr 1-4 is modified with Q already before the splicing in mouse brain; galQ on mature tRNA^Tyr cannot be resolved in APB northern blot (left panel). Red arrows indicate Q, black arrows indicate unmodified or unresolved galQ, wt: wild type, Q1: *Qtrt1^-/-*, Q2: *Qtrt2^-/-*.

that mQTRT1/2 is less active towards pre-tRNA^Tyr in comparison to tRNA^Tyr in vitro.

Subsequently, we mutated the UGU recognition motif in the anticodon loop[32] of pre-tRNA^Tyr 1-4 and a second putative UGU recognition motif in the intron of this isodecoder tRNA to determine if only one or both Gs can be modified to Q. Only $G_{34}$ in pre-tRNA^Tyr 1-4, but not the G in UGU of the intron, could be efficiently modified by mQTRT1/2 in vitro (Fig. 2a). Swapping of the introns between pre-tRNA^Leu 2-1 and pre-tRNA^Tyr 1-4 (Fig. 2a) did not reverse the $Q_{34}$ modification levels (Fig. 2c). Lastly, we tested the activity of the purified mQTRT1/2 complex on native pre-tRNAs. We used bulk tRNA isolated from wild type and Q1 and Q2-knockout mESCs, as substrates for the recombinant mQTRT1/2 and measured q-incorporation levels in vitro. The enzymatic complex could efficiently modify pre-tRNA^Tyr 1-4 as well as mature tRNA^Tyr from the pool of unmodified tRNA molecules in Q1 and Q2 mESCs (Fig. 2d, e).

Altogether, these findings demonstrate that intron-containing pre-RNA^Tyr serves as substrate for the QTRT1/2 complex with a similar efficiency as mature tRNA in vitro and in vivo although the modification reaction reconstituted in vitro is not equally efficient for different forms of tRNA^Tyr.

## Structural and functional insights into pre-tRNA recognition by the mQTRT1/2 complex

Next, we sought to understand the recognition of pre-tRNA^Tyr by the QTRT1/2 complex at the molecular level. We used the purified mQTRT1/2 complex (Fig. 3a) to quantitatively measure the binding parameters of the observed interaction and to detect minor differences in binding affinities for different in vitro transcribed mouse tRNAs by MST analyses (Fig. 3b). We performed the binding studies first in absence of inhibitor 9-deazaguanine (9dG) and then in its presence, as it is known to stabilize QTRT1/2-tRNA complex formation in vitro[12]. In addition, we used EMSA analysis to qualitatively assess QTRT1/2-tRNA complex formation before preparing cryo-EM grids (Fig. 3c). Even if the binding to pre-tRNA^Tyr 2-1 seems slightly weaker in MST assays, all tested tRNAs bind to the mQTRT1/2 complex with similar affinities (Fig. 3b, c, Supplementary Fig. 5a–d), showing that mouse QTRT1/2 is able to recognize intron-containing pre-tRNAs and that it does not counter-select for non-substrate tRNAs during the initial binding step, when we used tRNA^Leu as non-target negative control (Supplementary Fig. 5c). Moreover, mQTRT1/2 is also able to bind the mouse pre-tRNA^Tyr 1-4 that contains a 5′-leader and a 3′-trailer sequences (Supplementary Fig. 5c), which have been reported to be removed already in the nucleus[33]. Nonetheless, we do not have any in vivo evidence that QTRT1/2 acts even before these initial processing steps by RNase P, RNase Z, and CCA-adding enzymes.

Next, we prepared cryo-EM grids with the purified mQTRT1/2 (Fig. 3a) in the presence of 9dG and either mature tRNA^Tyr or the pre-tRNA^Tyr 1-4. Both types of samples were vitrified and after iterative rounds of grid optimization and screening, two single particle cryo-EM

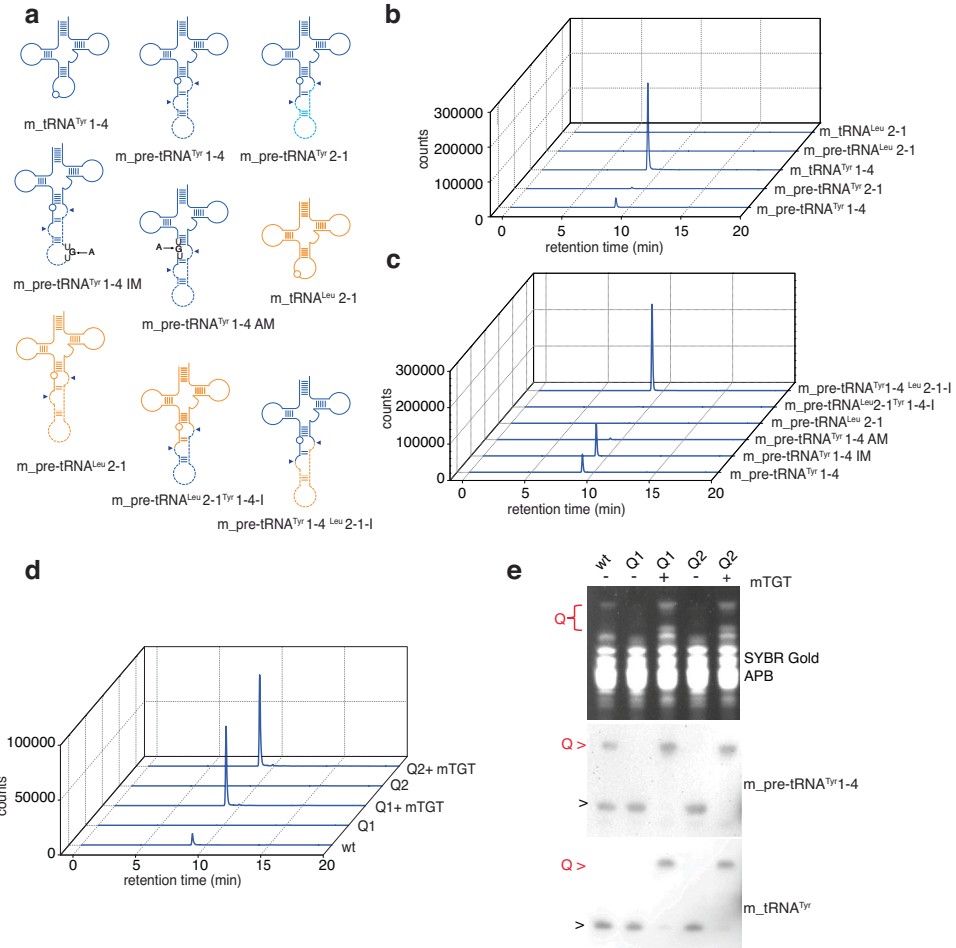

**Fig. 2 | mQTRT1/2 complex recognizes and modifies in-vitro transcribed and native pre-tRNA$^{Tyr}$. a** Schematic representation of in-vitro transcribed RNA according to the canonical cloverleaf structure of pre-tRNA and tRNA used in combination of mQTRT1/2 complex to assess the enzymatic specificity. The circles represent position 34, the triangles indicate the splicing sites. IM: intron mutant. AM: anticodon mutant. I: intron replacement. **b, c** The activity of the mQTRT1/2 complex is evaluated by LC-MS/MS on the same amount of injected RNA, measuring the incorporated queuine using the indicated in-vitro pre- and mature tRNAs after in vitro Q modification. tRNA$^{Leu}$/pre-tRNA$^{Leu}$ and tRNA$^{Tyr}$ are used as negative and positive control, respectively. mQTRT1/2 complex 200 nM, queuine 5 μM, 3 h at 37 °C. **d** The activity of mQTRT1/2 complex on bulk tRNA isolated from Q1 or Q2 mESCs is evaluated by LC-MS/MS, measuring the incorporated queuine after in vitro Q assay. Q1, Q2 and wt tRNAs are used as negative and positive control, respectively. **e** APB Northern blot of the samples indicated in **d** showing the activity of recombinant mQTRT1/2 complex (mTGT) on native pre-tRNA$^{Tyr}$ 1-4. Red arrows indicate Q, black arrows indicate unmodified or unresolved galQ, wt: wild type, Q1: *Qtrt1$^{-/-}$*, Q2: *Qtrt2$^{-/-}$*.

datasets were collected on a Titan Krios microscope equipped with a K3 direct electron detector. Following data collection, data curation, data processing and particle classification, a 3D reconstruction for both structures was obtained (Supplementary Fig. 6). After additional 3D classification and refinement routines, final maps at an overall resolution of 2.9 Å (mQTRT1/2-tRNA$^{Tyr}$) and 3.1 Å (mQTRT1/2-pre-tRNA$^{Tyr}$ 1-4), respectively, were obtained (Supplementary Table 1). The cryo-EM structure of mouse QTRT1/2 reveals the heterodimeric organization of the complex accommodating tRNA (Fig. 3d) that resembles the previously described apo crystal structures of mouse and human QTRT1/2 in complex with tRNA$^{Asp}$ [12]. Foremost, the structures show that mouse QTRT1/2 binds tRNA$^{Tyr}$ and the intron-containing pre-tRNA$^{Tyr}$ 1-4 in an almost identical fashion as human QTRT1/2 binds tRNA$^{Asp}$ [12] (Fig. 3e). QTRT1 binds and rearranges the anticodon stem loop (ASL) of the mature as well as the pre-tRNA to accommodate the modified position 34 in the active site. QTRT2 contributes to tRNA binding by contacting the D-loop and the acceptor stem. In the pre-tRNA structure, the initial residue of the intron can be located and at higher map thresholds and an additional density becomes visible at the expected location of the intron. However, the intronic sequence of pre-tRNA$^{Tyr}$ 1-4 seems to remain flexible and does not form a stable

secondary structure. The 9dG molecule is visible in the active sites of both structures, which show an overall r.m.s.d.$_{698\ residues}$ of 1.13 Å.

In summary, our cryo-EM structures confirm that mouse QTRT1/2 is able to accommodate both mature as well as precursor tRNA$^{Tyr}$ and that G$_{34}$ is positioned correctly in the active site to accomplish the G-to-Q exchange reaction, regardless of the presence of the intronic sequence.

## Galactose is further added to Q on pre-tRNA$^{Tyr}$ containing introns

Our in vitro and in vivo findings suggest that a subset of tRNA$^{Tyr}$ is modified with Q at the precursor stage. We could not identify any specific motifs that would distinguish the pre-tRNA$^{Tyr}$ isoforms that are modified in vivo from those that are not modified. In addition, structural models of the different intron-containing pre-tRNA isoforms bound to the mouse QTRT1/2 complex (Supplementary Fig. 7) using AlphaFold 3 show similar binding modes [34]. However, the highly expressed pre-tRNA$^{Tyr}$ 1-2 (Fig. 1d and Supplementary Fig. 1b), showed no shift in our APB Northern blot (Fig. 1f, g) indicating the lack of Q-modification in this abundantly expressed pre-tRNA. Yet, the presence of an additional modification on Q$_{34}$ can eliminate its capacity to

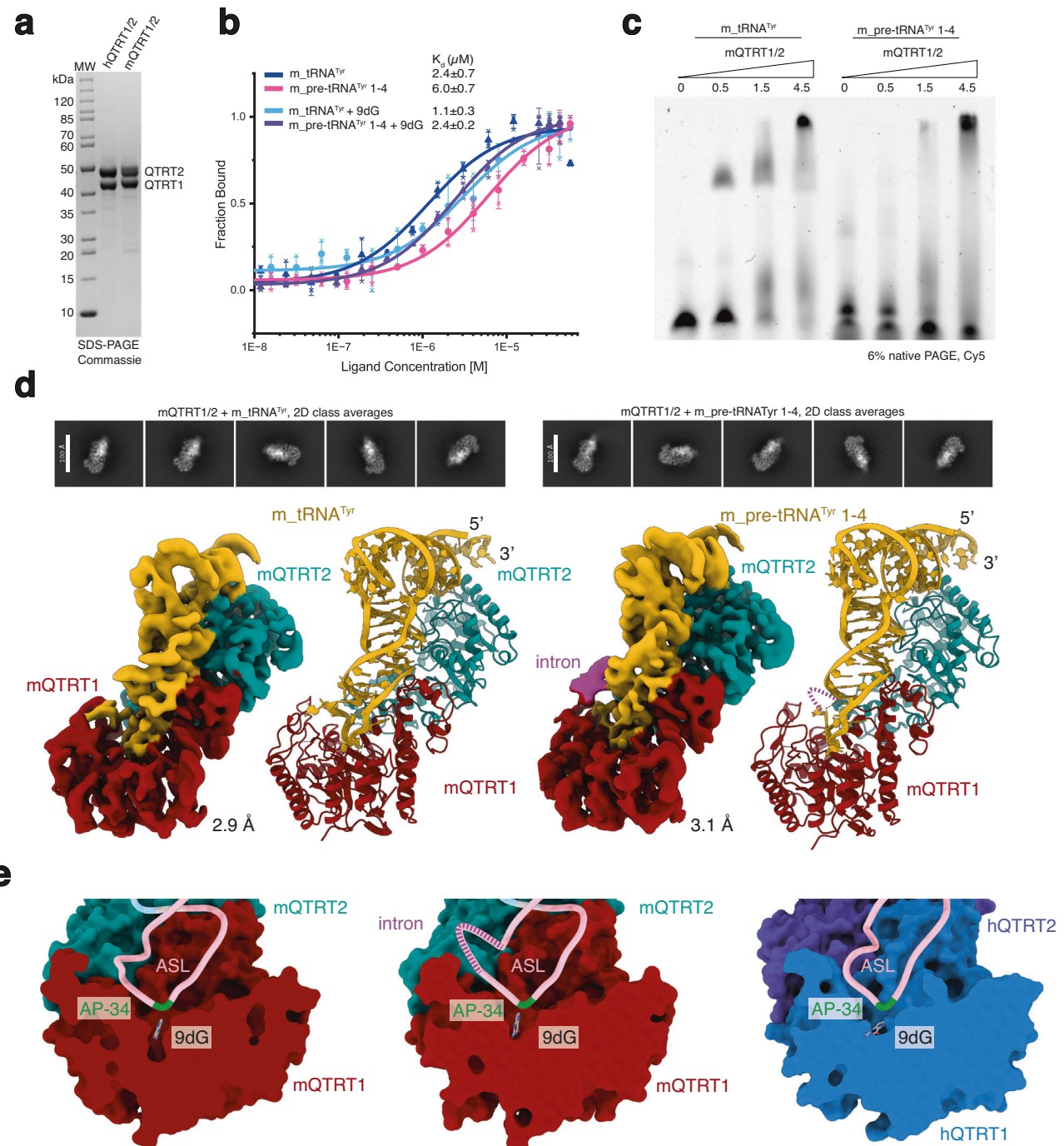

**Fig. 3 | Cryo-EM structures of mQTRT1/2 with tRNA^Tyr and pre-tRNA^Tyr 1-4.**
**a** Representative SDS-PAGE analysis of purified human and mouse QTRT1/2 complexes. MW: molecular weight marker. **b** MST analyses of mQTRT1/2 binding to tRNA^Tyr and pre-tRNA^Tyr 1-4. Concentration of mQTRT1/2 and calculated $K_d$ values ($K_d$ values) are given, concentration of tRNA is kept constant at 50 nM. Individual measurements at each concentration are shown with the fitted curve. Error bars represent ±SD ($n = 3$, biological replicates). **c** EMSA analyses of mQTRT1/2 binding to Cy5-labeled tRNA^Tyr and pre-tRNA^Tyr 1-4. Protein concentrations are given,

concentration of tRNA is kept constant at 200 nM. **d** Overview of structures of mQTRT1/2 complexes with tRNA^Tyr (left) or pre-tRNA^Tyr 1-4 (right). Representative 2D class averages (top) and cryo-EM maps are shown next to the respective atomic models (bottom). **e** Close-up view on the anticodon stem loop (ASL) region bound to the QTRT1 active site in mQTRT1/2 with tRNA^Tyr (left), mQTRT1/2 with pre-tRNA^Tyr 1-4 (middle) and hQTRT1/2 with tRNA^Asp (PDB ID 8OMR; right). Anticodon loop $G_{34}$ (AP-34) is in green, 9-deazaguanine (9dG) is visible in the active sites.

interact with boronic acid derivative in the APB-PAGE[35,36] (Fig. 4a). To test these hypotheses, we performed acid denaturing (AD) Northern blotting, a method recently used to detect glycosylated Q on tRNA[31]. Using this assay, we could confirm the presence of Q-containing modifications on mature tRNA^Tyr and pre-tRNA^Tyr 1-4 (Fig. 4b) and we could identify a hyper-modification also on pre-tRNA^Tyr 1-2 in wild type

mESCs, which was absent in Q1 knockout cells (Fig. 4b). Since we suspected that the modification is presumably galQ[13], we hypothesized that the additional sugar moiety on Q could potentially be removed enzymatically. Therefore, we digested small RNA purified from wild type and Q1 mESCs and mouse brains with a recombinant bovine β1-3,4 galactosidase (β-GAL) (Fig. 4a). Although the known substrates for this

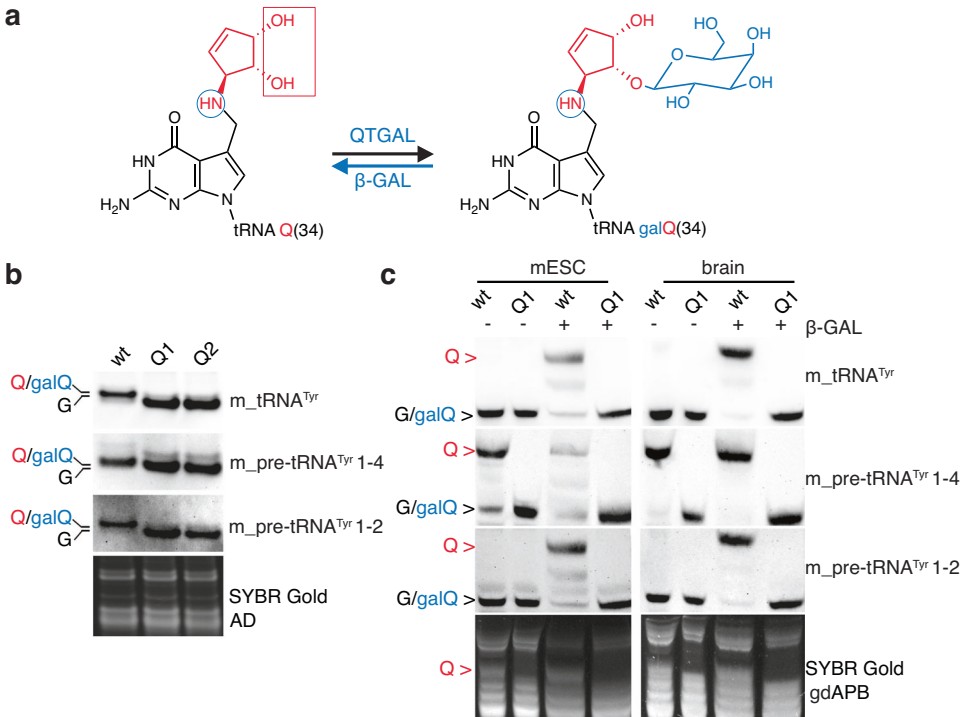

**Fig. 4 | pre-tRNA^Tyr 1-2 is modified at G_34 with galQ before splicing. a** The presence of galactose modification on $Q_{34}$ can eliminate its capacity to interact with boronic acid derivative in the ABP-PAGE. β1-3,4 Galactosidase (β-GAL) is used to hydrolyze terminal β-galactose residues from the precursor and mature tRNA^Tyr. Blue circles indicate the secondary amine protonated and reacting in AD Northern blot. Red rectangle indicates the cis-diol group reacting in APB Northern. **b** Both pre-tRNA^Tyr1-4 and 1-2 are shifted in wild type mESCs using AD Northern indicating the presence of a Q derivative modification. **c** Digestion of galQ using β1-3,4 Galactosidase (β-GAL) revealed the Q derivative modification on mature m_tRNA^Tyr and m_pre-tRNA^Tyr1-2 in mESCs and the mouse brain using gdAPB Northern blot.

enzyme are various oligosaccharides and glycopeptides, we could efficiently remove galactose from Q on tRNA and detect Q on pre-tRNA^Tyr 1-2 by APB Northern blotting and mass spectrometry (Fig. 4c and Supplementary Fig. 8a–c). Using this approach, which we call gdAPB (glycosidic digestion acryloyl aminophenyl boronic acid-based) Northern blotting we were able to show that ~85% (in mESC) and ~98% (in mouse brain) of pre-tRNA^Tyr 1-2 are queuosinylated and further galactosylated (Fig. 4c and Supplementary Fig. 8a–c) and as expected no manQ can be detected on mature tRNA^Tyr and on the tested pre-tRNAs (Supplementary Fig. 8d). Our method is further validated by the detection of galQ_34 in mature tRNA. As previously described[37], we observed that ~80% and ~95% of mature tRNA^Tyr is galQ_34-modified in mESCs and mouse brains, respectively (Fig. 4c). This new approach, which combines the enzymatic digestion of the galQ_34 hypermodification with the APB affinity electrophoresis, allowed us to efficiently discriminate between Q and galQ modifications at single isodecoder resolution and demonstrated that a subset of mouse pre-tRNA^Tyr is already galQ_34-modified before the splicing.

### Pre-tRNA^Tyr is modified in several eukaryotes

The TGT complex as well as glycosylation enzymes are broadly conserved in eukaryotes[13]. Hence, we were interested to examine the modification status of precursor tRNA^Tyr in other organisms, including humans and model organisms including *D. melanogaster* and *C. elegans*. In the human genome, 13 different genes for tRNA^Tyr can be identified, all of which differ in their intronic sequences (Supplementary Fig. 9). We used structural predictions to show that in principle all of these pre-tRNA could be bound by the human QTRT1/2, while accommodating G_34 in the active site (Supplementary Fig. 10). Our systematic analysis in cervical carcinoma HeLa cell line grown in a medium supplemented with dialyzed serum, which

deprives Q from tRNA[30], in presence or absence of q, identified galQ_34 modification on the unspliced forms of human pre-tRNA^Tyr 8-1 and pre-tRNA^Tyr 9-1 (Fig. 5a) as well as Q on pre-tRNA^Tyr 3-1, pre-tRNA^Tyr 5-3 and pre-tRNA^Tyr 5-5 (Supplementary Fig. 9b).

Next, we tested the presence of Q-modifications in flies, which apparently lack an ortholog for QTGAL and hence only have Q- but no galQ-modified tRNAs. We adapted *D. melanogaster* S2 cells with medium supplemented with dialyzed serum in the presence or absence of q (Fig. 5b) and analyzed pre-tRNA^Tyr isodecoders by ABP Northern blots. Thereby, we identified pre-tRNA^Tyr 1-2, pre-tRNA^Tyr 1-4, pre-tRNA^Tyr 1-6, pre-tRNA^Tyr 1-7, and pre-tRNA^Tyr 1-8 to be queuosinylated before splicing (Fig. 5b, Supplementary Fig. 11a, b). Q modification of the most abundant pre-tRNA^Tyr 1-4 could be also be confirmed in adult flies (Supplementary Fig. 11c).

Lastly, we looked at the 18 tRNA^Tyr isodecoders in the *C. elegans* genome, nine of which have a different intron sequence (Supplementary Fig. 11d). We grew the N2 wild type strain of *C. elegans* in plates with very low or high peptone, which is known to modulate the intracellular levels of Q_34−modified tRNA (Supplementary Fig. 11e) and standard *E. coli* OP50 bacterial strain. In *C. elegans*, queuosinylation of mature tRNA^Tyr was found to be weak, and we could identify only pre-tRNA^Tyr 2-8 to be modified with Q (Fig. 5c). Even if the QTGAL gene is present in the worm genome, our Northern blots did not reveal galQ-modified pre-tRNA^Tyr (Fig. 5c and Supplementary Fig. 11f).

By modulating Q_34 levels in cells of different organisms and in live worms, we could demonstrate that pre-tRNA^Tyr is not only an in vivo substrate for the QTRT1/2 complex, but also for QTGAL in different species (Supplementary Table 2). Considering the various organisms, representing distinct clades of eukaryotic evolution addressed in this study, we suggest that queuosinylation of pre-tRNA is a conserved feature of the tRNA maturation process across most eukaryotes.

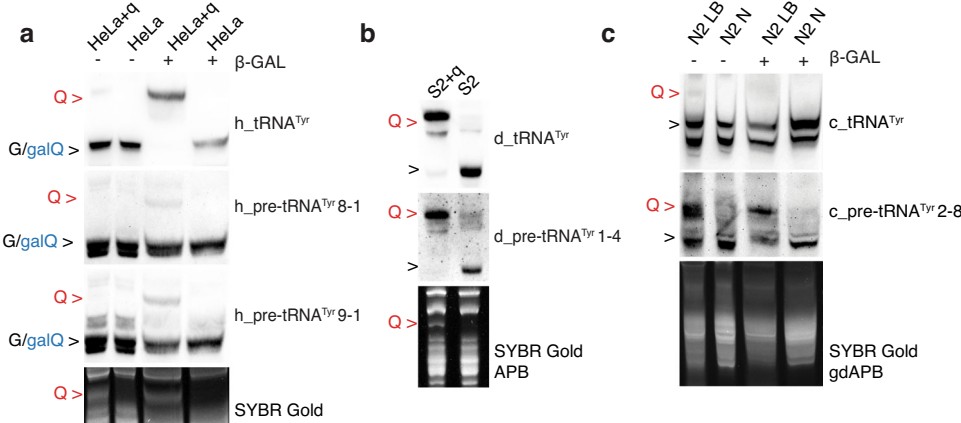

**Fig. 5 | Evolutionary analysis of the pre-tRNA$^{Tyr}$ modification at position 34 using Northern blot. a** galQ modification is detected on mature h_tRNA$^{Tyr}$ and on h_pre-tRNA$^{Tyr}$ 8-1 and 9-1 in human HeLa cells grown in medium ± q using gdAPB Northern blot. **b** *D. melanogaster* is missing QTGAL enzyme and Q modification is detected on d_pre-tRNA$^{Tyr}$ 1-4 in S2 cells grown in medium ± q using APB Northern

blot. **c** Q modification is detected on mature c_tRNA$^{Tyr}$ and on c_pre-tRNA$^{Tyr}$ 2-8 in N2 wild type strain of *C. elegans* grown in peptone rich medium (LB) versus NGM plates (N) using gdAPB Northern blot. Red arrows indicate Q, black arrows indicate unmodified or unresolved galQ.

## Discussion

Precursor tRNA molecules, transcribed in the nucleus by RNA Polymerase III[38] must undergo an elaborate set of post-transcriptional processing steps to generate mature, fully functional tRNAs competent in decoding mRNAs during protein synthesis at the ribosome[5,39]. Although we know that mature tRNAs carry numerous nucleotide modifications that are introduced post-transcriptionally[40,41], it mostly remains unclear at which processing stage each respective modification is introduced[42]. Here we study this intricate sequence of events in vivo and in vitro, showing that Q and galQ, a hyper-modification derived from Q, are added to certain isodecoders of tRNA$^{Tyr}$ at the precursor stage, to molecules that still contain the intron, thus occurring before splicing. We confirm this order of events in human and several eukaryotic model organisms, including mice, flies, and worms.

The relative order of biological processes is often governed by the specific sub-cellular localization of the responsible enzymes. However, the dynamic relocation of maturing pre-tRNA molecules between different compartments, as well as inconsistent reports about the location of processing and modifying enzymes, complicates the interpretation for the tRNA maturation process. In yeast, for instance, the tRNA splicing machinery is found on the surface of mitochondria[33,43], whereas the cytoplasmic localization of the human TSEN complex remains controversially discussed[44]. Initial reports rather pointed towards a nuclear localization[45,46], but more recent studies show a membrane-bound or cytoplasmic positioning of the splicing machinery[44,47,48]. Intron-containing pre-tRNAs were found to be enriched in the cytoplasm of human cells[44,49], which suggests that they are also spliced in the cytoplasm. As the human QTRT1/2 complex itself is found also on the mitochondrial surface[50], it is possible that it modifies intron-containing pre-tRNA at that cellular location.

Pioneering work obtained by the injection of yeast pre-tRNA$^{Tyr}$ into *Xenopus laevis* oocyte nuclei showed that Q$_{34}$ is introduced in the anticodon loop only after splicing, in tRNAs exported to the cytoplasm[51]. However, Q is normally absent in yeast tRNAs[13] and oocytes hardly transcribe and/or process tRNAs. It is also not surprising that in *T. brucei* Q-incorporation is observed post-splicing, since this organism has evolved several distinct mechanisms for production and processing of the RNAs, which sets it apart from animals and fungi[52]. In detail, pre-tRNA$^{Tyr}$ from *T. brucei* is first exported from the nucleus, spliced in the cytoplasm and

subsequently transported back to the nucleus via retrograde transport, where it undergoes Q$_{34}$ modification[19]. Subsequently, the Q-modified tRNA needs to be re-exported to the cytoplasm for translation, with a fraction being further selectively imported into mitochondria[53,54]. Our works aims to clarify at what maturation stage Q-modifications, namely Q and galQ, are introduced in metazoans including humans, where also similar complex and dynamic tRNA trafficking happens between the nucleus, the cytoplasm and the mitochondrial surface[33].

Of note, certain other tRNA modifications have been described to occur on pre-tRNAs before splicing. For instance, yeast Pus7 converts uridine into pseudouridine (Ψ) in pre-tRNA$^{Tyr}$ at position 35 and human PUS3 is able to bind and modify pre-tRNA$^{Arg}$ at position 39 in vitro[55,56]. Currently, it is not known whether Ψ$_{35}$ is added on pre-tRNA$^{Tyr}$ before or after Q$_{34}$. The mainly nuclear localization of PUS enzymes[57] suggests that Ψ comes first, but no functional link between Ψ$_{35}$ and Q$_{34}$ has been reported so far. We could not identify any sequence motifs or secondary structures that discriminate tRNA$^{Tyr}$ precursors that remain unmodified from those that receive Q$_{34}$ or are further modified to galQ$_{34}$. We cannot exclude that the presence or absence of other tRNA modifications contributes to these differences. It is well known that tRNA trailer and leader sequences are removed already in the nucleus[33,58], nevertheless, we observed a consistent binding affinity of QTRT1/2 for a pre-tRNA$^{Tyr}$ that still contains those extensions, therefore we cannot exclude that Q is added very early in the processing. Furthermore, mature tRNA$^{Tyr}$ coming from those precursors, which were never observed to be modified (Supplementary Table 2), might still undergo modification after being spliced. Our results demonstrate that the presence of intronic sequences does not influence the deformation of the ASL region, imposed upon binding to QTRT1/2 complex. In detail, this structural rearrangement of tRNA, positions G$_{34}$ in the reaction center, results in a similar geometry for both mature- and pre-tRNA, suggesting that intronic sequences that would interfere with the modification reaction have been counter-selected throughout evolution. Therefore, other unknown factors may be responsible for this specificity, especially regarding the discrimination of pre-tRNA$^{Tyr}$ 1-2 and 1-4 by QTGAL, since the first is galactosylated and the second only queuosinylated in mouse in vivo. Those varieties may be caused by different levels of expression and cellular localization of various precursors or yet unknown factors.

The hierarchical interplay between tRNA splicing and modification order remains elusive in the functional sense and evolutionary impact. This is an unsolved fascinating question and one may speculate that this is an important process for quality control steps during tRNA maturation. In this regard, we observed that the absence of Q leads to an increase of intron-containing pre-tRNA$^{Tyr}$ fragments in mouse tissues and neuronally differentiated cells. These aberrant fragments could indicate that Q modification is necessary for the stability of the pre-tRNA, similar to what was observed for mature tRNA and as previously described for cancer cells[59], or for preventing defects in the splicing mechanism[24]. Indeed, dysregulated tRNA processing and the accumulation of potentially toxic tRNA fragments have been indicated as a cause of pontocerebellar hypoplasia[25,60]. This disease is characterized by neurodegeneration resulting in the atrophy of the cerebellum and other parts of the brain during development. Mutations in all four subunits of the TSEN complex and in CLP1 have been reported as the cause of pathology, which displays a variety of symptoms including microcephaly, loss of motor development and intellectual disabilities[24,61]. A recent study in yeast proposed that free tRNA introns may play a role in regulating post-transcriptional gene expression[62], however, how and if pre-tRNA fragments may contribute to the onset and progression of the neuronal degeneration is still under study[44]. Interestingly, QTRT1 knockout mice display a reduction of hippocampal cytoarchitecture resembling atrophy, with learning and memory defects especially in the female Q1 mice[15], suggesting a similar mechanism of pathogenesis. As the lack of Q does not influence mature tRNA$^{Tyr}$ levels (Supplementary Fig. 2)[15], alterations in the Q-modification pathway most likely will not affect any additional biological activities that might be carried out by the produced intronic sequences.

In summary, the presented results reveal the presence of Q and galQ modifications in pre-tRNAs containing introns in cells from various eukaryotic organisms. Our work also provides a mechanistic explanation of how the intron-containing tRNAs are recognized and modified by the QTRT1/2 complex. Last but not least, our study provides insights into the highly dynamic, physiological processes that are relevant for pathological pathways in patients.

# Methods

## Cell culture
E14 mouse ESCs were cultured on 0.2% gelatin-coated plates with Knockout DMEM medium (Gibco), supplemented with 10% FBS (Gibco), 1% GlutaMax (Gibco), 1% penicillin/streptomycin (Gibco), 100 μM beta-mercaptoethanol (Sigma-Aldrich), and $1.2 \times 10^3$ U LIF (ESGRO). Qtrt1 (Q1) and Qtrt2 (Q2) mESCs knockout clones were previously described[63]. Human HeLa cells were cultured with complete DMEM medium (Gibco), supplemented with dialyzed FBS (Sigma). *D. melanogaster* S2 cells were cultured with complete Schneider's Drosophila Medium (Gibco), supplemented with dialyzed FBS. 20 nM queuine was added to the medium in all +q conditions.

## Animal handling
The husbandry of mice was performed at Mannheim Faculty of Medicine, University of Heidelberg with 12:12 light:dark cycles at standard housing temperatures of 18–23 °C. The animal tissue dissection was carried out in strict compliance with national and international guidelines for the Care and Use of Laboratory Animals (Regierungspräsidium Karlsruhe, Germany). The experiments were performed on 2-month-old littermates wild type and C57BL/6J-Qtrt1$^{emITuo}$ homozygous (Q1) mice[15]. *C. elegans* N2 strain was grown on NGM as already described[64], or alternatively on LB plates to increase the peptone supply. *E. coli* OP50 strain was used as the food source (Caenorhabditis Genetics Center, University of Minnesota, Twin Cities, MN, USA). *D. melanogaster* w$^{1118}$ strain was raised on standard

media and maintained in incubators with controlled temperature and humidity on a 12 h light/dark cycle.

## RNA isolation
Total RNA was isolated from $10^5$-$10^7$ cells or ~50 mg tissue using 1 mL of TRIzol (Invitrogen), according to the manufacturer's instructions. The concentration and purity of the isolated RNA were measured using NanoDrop and the samples were stored at −80 °C.

## APB, PAGE, AD, and gdAPB Northern blotting
APB Northern blotting is based on the co-polymerization of *N*-acryloyl-3-aminophenylboronic acid (APB), in polyacrylamide gels, and on the interplay between the boronate moiety and free cis-diol groups of the Q-tRNA (Fig. 1a). Denaturing polyacrylamide gel electrophoresis (PAGE) and *N*-acryloyl-3-aminophenylboronic acid (APB) gel electrophoresis were performed as described before[15]. 10 μg of total RNA was loaded for each well. The AD Northern blotting was performed as described previously[31]. For gdAPB Northern blotting, 1 μg of small RNA (17-200 nucleotides) was purified using the RNA Clean and Concentrator kit (Zymo) from total RNA according to the manufacturer's instructions. The isolated small RNA was treated with 8 U of β1-3,4 galactosidase (NEB) or with 2 U of α1-2,3,6 mannosidase (NEB) at 37 °C for 60 min; the samples were then deacylated, denatured, and run as described for APB Northern blots previously. Northern blot membranes and gels were scanned using ChemiDoc MP Imaging System (BioRad) and analyzed using Adobe Photoshop 22.2.0. Probes used in Northern blots are listed in Supplementary Data 1 and as previously described[15,29,30].

## In vitro tRNA transcription
The tRNA was produced using the T7 RNA polymerase-mediated run-off method[65]. The in vitro transcription reaction was performed in a 500 μL volume containing DNA template, T7 RNA polymerase, and reaction buffer (20 mM Tris, pH 8.0, 5 mM DTT, 150 mM NaCl, 8 mM MgCl$_2$, 2 mM spermidine, 20 mM NTPs, RNasin, and pyrophosphatase). The reaction was performed at 37 °C overnight and followed by DNase I treatment to remove DNA templates. The product was then purified using a DEAE column and heat treatment at 80 °C for 2 min and followed by the slow cooling process to room temperature as the re-annealing process. To obtain a homogenous tRNA population, the samples were subjected to a Superdex 75 Increase gel filtration column, and the tRNA-containing fractions were pooled and stored at −80 °C. Alternatively, DNA templates, designed to carry a T7 polymerase promoter, were amplified via PCR. In vitro transcription was performed using HiScribe T7 High Yield RNA Synthesis Kit (NEB) following the supplier's protocol with 1 μg of respective DNA template and 20% (V/V) DMSO (Carl Roth) in a total volume of 25 μL. Incubation was performed overnight at 37 °C. To remove the DNA template, 10 U of DNase I-XT (NEB) as well as the respective reaction buffer was added to crude IVT mixture and incubated in a total volume of 100 μL for 1 h at 37 °C. Purification was performed using Monarch® Spin RNA Cleanup Kit 500 μg (NEB). To remove side products, the IVT mixture was applied onto a 10% denaturing polyacrylamide gel. Each band was cut and crushed, then the tRNA was purified by filtering through NanoSep® centrifugal filters (PALL) and was subsequently ethanol precipitated. The DNA template contained a T7-promoter sequence followed by the tRNA sequences listed in Supplementary Data 1.

## TGT recombinant complex
Expression and purification of murine TGT complex and the determination of Michaelis–Menten parameters were carried out as previously described[11]. In short, the murine QTRT1 and QTRT2 were co-expressed in *Vibrio natriegens* Vmax cells in 2 x YT medium. After induction at an OD600 of 1.0 with 1 mM IPTG, expression occurred for 24 h at 15 °C.

After centrifugation, the cells were resuspended in lysis buffer (100 mM Tris pH 7.5, 150 mM NaCl, 1 mM Na-EDTA), lysed via ultrasound four times for 4 min and purified on a Strep-Tactin XT 4Flow® high capacity FPLC column (IBA Lifescience). To remove nucleic acid impurities, the column was washed with a lithium-containing buffer (lysis buffer + 750 mM LiCl) before elution and subsequently, after cleavage of the affinity tags with PreScission protease and removal of the protease by a GST-column, the protein was purified on a phenyl-sepharose-column (Phenyl High performance 16/10, GE Healthcare). The elution was performed with Phenyl Sepharose elution buffer (15 mM Tris pH 7.5, 25 mM NaCl) using a linear gradient increasing the buffer concentration from 0 to 100% in 35 mL with a flow rate of 3 mL/min. For usage in the assay, the protein was dialyzed against assay buffer (100 mM HEPES pH 7.3, 20 mM MgCl$_2$, 0.0085% Tween20) and adjusted to a concentration of 300 nM.

### In vitro Q modification with recombinant TGT

3 µM of in vitro transcribed tRNA or alternatively 1 µg of small RNA isolated from mESCs, was incubated with 200 nM murine QTRT1/2 complex in TGT reaction buffer containing 50 mM Tris pH 7.5, 20 mM NaCl, 5 mM MgCl$_2$, 2 mM dithiothreitol (all Carl Roth) and 10 µM queuine (provided by Reuter lab[66]) to a total volume of 30 µL. For proper folding of the tRNAs, reaction was set in following order: tRNAs and the needed amount of purified water were mixed and heated to 75 °C for 2 min followed by a stepwise reduction of 5 °C every 30 sec until 40 °C, then TGT reaction buffer was added and the reaction was allowed to cool down before adding the other components. The incubation was performed for 3 h at 37 °C. Reaction mixture was purified using RNA Clean & Concentrator-5 (Zymo) according to the supplier's protocol and elution was performed with 6 µL purified water.

### Liquid chromatography–tandem mass spectrometry (LC−MS/MS)

For LC-MS/MS analysis either 15 pmol of in vitro treated IVT or 700 ng of in vitro treated tRNA from Q1 or Q2 knockout or untreated tRNA from wild type mESCs, or 400 ng of β-gal treated or untreated tRNA from HeLa cells or mice were digested in a total volume of 20 µL using an enzyme mixture containing 10 U benzonase (Sigma), 0.2 U snake venom phosphodiesterase from *C. adamanteus* (Worthington), 0.2 U bovine intestine phosphatase (Sigma), 200 ng Pentostatin (Sigma) and 0.6 U nuclease P1 from *P. citrinum* (Sigma) in a buffer containing 5 mM Tris (pH 8) and 1 mM MgCl$_2$. Reaction was performed for two hours at 37 °C and quenched by diluting to 50 µL. Analysis was executed on an Agilent 1260 series LC with an Agilent 6470B Triple Quadrupole mass spectrometer containing an electrospray ion source (gas temperature 300 °C, gas flow 7 L/min, nebulizer pressure 60 psi, sheath gas temperature 400 °C, sheath gas flow 12 L/min, capillary voltage 3000 V). Separation was performed using a Synergi Fusion RP18 column (250 × 2.0 mm, 4 µM particle size, 80 Å pore size; Phenomenex) with two solvents. Solvent A consisted of 5 mM ammonium acetate (Merck) adjusted to pH 5.3 using acetic acid (Honeywell Chemicals) while buffer B was LC-MS grade acetonitrile (VWR Chemicals) with the addition of 1% (V/V) purified water. The gradient started with 100% solvent A followed by a linear increase of solvent B to 8% within 10 min, which was raised to 40% solvent B after 20 min. Initial conditions were restored within 3 min and held for an additional 10 min. The flow rate was constant at 0.35 mL/min. UV traces at 254 nm were registered using a diode array detector. Q, or galQ and manQ were detected using multiple reaction monitoring (MRM) in positive mode (mass transitions: Q 410 - > 295 and 410 - > 163, galQ and manQ 572 - > 163 and 572 - > 295).

### Microscale thermophoresis (MST)

In this assay, we used tRNA (labeled with Cy5-cytosine on the stage of in vitro transcription). Murine QTRT1/2 complex purified as described above was serially diluted in the constant (50 nM) concentration of respective, Cy5-labeled tRNA and buffer containing 20 mM HEPES, pH 7.5, 75 mM NaCl, 1 mM MgCl$_2$, 0.025% Tween 20 (NanoTemper Technologies) and 2 mM DTT. To derive apparent $K_d$ values in the presence of 9dG, inhibitor was added to the samples in a final concentration of 100 µM. Samples were incubated at 25 °C for 30 min and then loaded into premium capillaries (NanoTemper Technologies). Measurements were performed using the Monolith device (NanoTemper Technologies) and data were analyzed using MO. Control software (NanoTemper Technologies) and MO. AffinityAnalysis (Nano Temper Technologies). $K_d$ (or apparent $K_d$) values were calculated based on three independent experiments.

### Electrophoretic Mobility Shift Assay (EMSA)

In this assay, we used tRNA labeled with Cy5-cytosine on the stage of in vitro transcription. Murine QTRT1/2 complex purified as described above was prepared in three concentrations per experiment (5.0; 1.5; 0.5 µM) in the constant (200 nM) concentration of respective, Cy5-labeled tRNA and buffer containing 20 mM HEPES, pH 7.5, 75 mM NaCl, 1 mM MgCl$_2$, 2 mM DTT and 100 µM 9dG. Samples were incubated at 4 °C for 10 min. mQTRT1/2-tRNA complex and free tRNA were resolved using 6% native gel, by running at 100 V, 4 °C for 3 h. Gels were scanned using ChemiDoc MP Imaging System (BioRad) and analyzed in Image Lab Software (BioRad).

### Cryo-EM grid preparation

Murine QTRT1/2 complex purified as described above was mixed with respective, in vitro transcribed tRNA$^{Tyr}$ (with and without intron) in 1:1.5 molar ratio (in the EMSA buffer), and incubated at 25 °C for 30 min. Immediately after the incubation, 3 µL of the respective sample was applied onto glow-discharged CryoEM grid (Quantifoil R2/1, Cu 200 mesh) and plunge-frozen into liquid ethane using Vitrobot Mark IV (Thermo Fisher Scientific) in the following conditions: 100% humidity, 4 °C, blot time: 2 s, blot force: 5, wait time: 1 s.

### Cryo-EM single-particle reconstruction

Datasets were collected using 300 keV Tiran Krios G3i (Thermo Fisher, Solaris, Poland) equipped with Gatan BioQuantum energy filter and a K3 direct electron detector. Micrographs were acquired using pixel size of 0.86 Å in the defocus range of -0.6 to -1.5 over 40 frames accumulating a total dose of 40 e-/Å$^2$. Initial data processing was performed in CryoSPARC 4.2.0[67–69]. Motion-corrected, averaged micrographs were subsequently corrected using CTF. Then, particles were picked using blob-picker, extracted and subjected into 2D classification. Obtained 2D class averages served as templates for template-picking. Template-picked particles were extracted and subjected to iterative 2D classification. Ab initio reconstruction was run on the best set of particles. The initial model was used for a non-uniform refinement with all available, high-quality particles. Particles with assigned angular information were transferred to Relion[70] for iterative 3D classification. The best 3D classes were combined and subjected to final refinement.

### Molecular modeling

Atomic structures of mQTRT1/2-tRNA$^{Tyr}$ and mQTRT1/2-pre-tRNA$^{Tyr}$ 1-4 were assembled based on mQTRT apo crystal structure (PDB: 7OV9) and respective tRNA models predicted with AlphaFold 3. Models were fitted into the experimental densities using ChimeraX (rigid-body fit)[71], then flexibly fitted using ISOLDE (maintaining all secondary structures) and manually curated using WinCoot[72]. Models were further refined and validated in Phenix.

**Statistics and reproducibility.** All experiments were conducted with at least two independent biological replicates, and a single representative image is shown for each Northern blot or gel. Sample sizes

were determined in advance based on standard practices described in the literature. No data were excluded from our analyses. Mice were randomly selected from both sexes and genotypes. For Cryo-EM data collection, we used EPU v.2.10.0.1941REL (Thermo Fisher). Data collection and analysis were not performed blind to experimental conditions. Investigators were not blinded to cellular or animal genotyping information. Diagrams and statistical analyses were performed using GraphPad Prism 8.4.3. Statistical significance is expressed as mean ± SD and was assessed using a two-tailed Student's $t$ test. $p$-values < 0.05 were considered statistically significant. Specific $p$-values are provided in the Source Data file. The diagrams shown in Figs. 1b and 2a were created with Adobe Illustrator 24.1.2. Chemical structures were designed using ChemDraw 20.0.

### Reporting summary

Further information on research design is available in the Nature Portfolio Reporting Summary linked to this article.

## Data availability

A reporting summary for this article is available as Supplementary Information file. All the data supporting the findings of this study are available within the article, its Supplementary Figures and within the Source Data file. The atomic models and cryo-EM densities maps have been deposited in the EMData Bank (EMDB) and the Protein Data Bank (PDB) under the following accession codes−mouse QTRT1/2 with mature tRNA$^{Tyr}$ (PDB ID 9HN7; EMD-52308) and mouse QTRT1/2 with precursor tRNA$^{Tyr}$ 1-4 (PDB ID 9HN9; EMD-52309). Uncropped and unprocessed scans of the gels and Northern blots are provided in the Source Data file. Source data are provided with this paper.

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

## Acknowledgements

The authors thank Georg Stoecklin and Frank Lyko for the valuable discussions and support. Aurelio Teleman for providing *Drosophila melanogaster w*[1118] strain. The transgenic facility at DKFZ for providing mouse E14 ESCs. We also thank Doris Linder and Thi Bach Nga Ly-Hartig for technical support. F.T. was supported by the Institute of Genetics and Biophysics A. Buzzati-Traverso, The National Research Council of Italy. This work was funded by grants from the Deutsche Forschungsgemeinschaft (DFG, German Research Foundation) TU5371-2 to F.T., project number 439669440 TRR319 RMaP TP C03 to M.H., A06 to F.T. and JP., and an Emmy Noether Programme grant (442512666) to JP. The work was further supported by the European Research Council (ERC) under the European Union's Horizon 2020 research and innovation program (101001394; S.G.). In addition, we thank the MCB SBCF (TEAMTECH_CORE_FACILITY/2017-4/6; FNP),

SOLARIS (Polish Ministry and Higher Education; 1/SOL/2021/2) and PLGrid ACK Cyfronet (PLG/2024/016848) for access. For the publication fee we acknowledge financial support by Heidelberg University and by DFG project number 439669440 TRR319 RMaP.

## Author contributions

F.T., M.H. and S.G. conceived the study. W.G., I.K., K.K., F.F., S.R. and L.K. designed, performed and visualized the experiments with the supervision of F.T., M.H., K.R., S.G. W.G., S.R. performed the animal and cellular handling and the Northern blot analyses. C.C. generated the mouse knockout cell lines. F.F. and K.R. provided the mouse recombinant QTRT1/2 complex. E.C. and J.P. provided the TSEN recombinant complex and performed the splicing assay. E.C., W.G., F.F., K.K. and I.K. performed the in vitro modification assays. K.K. performed the mass spectrometry analysis. I.K., A.C.G., L.K. performed the cryo-EM analysis. W.G., I.K., S.G. and F.T. wrote the manuscript with the contribution from the other coauthors.

## Funding

## Competing interests

The authors declare no competing interests.
