## [Transparent Peer Review file · Nature Communications]

Queuosine is incorporated into precursor tRNA before splicing

Corresponding Author: Dr Francesca Tuorto

Version 0:

Reviewer comments:

Reviewer #1

(Remarks to the Author)

In the manuscript "Queuosine is incorporated into precursor tRNA before splicing", Guo et al. investigate the temporal order of events in tRNA maturation, focusing on tRNA-splicing and Q-incorporation. The authors observe the presence of Q or galQ -modified pre-tRNAs containing introns, both in vitro and in vivo. The authors also demonstrate the similar recognition mechanisms of mouse QTRT1/2 in complex with mature tRNATyr and the intron-containing pre-tRNATyr 1-4 by single particle cryo-EM. Overall, this manuscript is well written and well structured. This study provides insights into the highly dynamic and physiological processes involved in tRNA maturation. However, several points need to be addressed before publication in Nature Communications.

In their study, the pre-tRNALeu served as a negative control, but interestingly pre-tRNALeu binds to the mQTRT1/2 complex with similar affinities as pre-tRNATyr1-4. To better understand the underlying mechanisms, it would be helpful to provide the AlphaFold 3 prediction of mQTRT1/2 in complexes with pre-tRNALeu, comparing this model with the complex formed with the intron-containing pre-tRNATyr 1-4, to see if there are any differences around active site between the binding modes.

The study shows that the presence of intronic sequences does not influence the Q modifications. The mQTRT1/2 complex recognize the exon regions of tRNAs, and the different intron-containing pre-tRNA isoforms bind to the mouse QTRT1/2 complex in similar binding modes (Fig.S6). However, many pre-tRNATyr isoforms were expressed but not modified (Table S2). The authors should clarify why certain pre-tRNATyr isoforms fail to undergo modification, despite having similar structural features.

In vertebrate tRNATyr, Q is further glycosylated with galactose and mannose to generate galQ and manQ, respectively. Is it possible that manQ is also added to pre-tRNATyr before the splicing of introns ?

Intron-dependent tRNA modification sites are typically modified prior to splicing. The authors show that Q is added to pre-tRNATyr before splicing, yet the activity of the mQTRT1/2 complex appears to be higher on mature tRNATyr (Fig. 2b). This raises the question of whether Queuosine modification occurs on both intron-containing and intron-lacking substrates, and if so, how these two forms of the substrate differ in terms of modification efficiency, and whether the presence or absence of Q modifications influences the splicing process.

Minor points:

- Please provide a close-up view of the electron density around the anticodon stem loop (ASL), including the intron region.
- Please add a reference to Fig.4a in the main text.
- Report min/mean/max B factors in Table S1

Reviewer #2

(Remarks to the Author)

The hypermodified nucleoside queuosine is introduced in tRNA^{His}, tRNA^{Tyr}, tRNA^{Asp}, and tRNA^{Asn} by the enzyme tRNA-guanine transglycosylase (QTRT1/QTRT2), which replaces the guanine-34 in the anticodon by queuine.

In this manuscript Guo, Kaczmarczyk and colleagues provide evidence that for tRNA^{Tyr} the Q-modification is formed before splicing. They analyzed tRNAs from mouse embryonic stem cells by means of ABP gels and northern blotting. Only a small fraction of below 4% of mature tRNA^{Tyr} contains Q, as it contains galQ, which cannot be detected by the ABP method. Interestingly, 60% of pre-tRNA^{Tyr} shows a signal on the ABP gel, which has been interpreted as Q-content. However, as stated by the authors, the ABP method is not specific for Q, it rather detects cis-diol groups, like in ribose.

The authors performed in vitro assays to demonstrate that pre-tRNA^{Tyr} is a substrate for TGT, and they also determined the cryo-EM structure of the pre-tRNA^{Tyr} - TGT complex. This structure closely resembles the tRNA-TGT structure, but the intron is not resolved in the EM reconstruction as it appears to be flexible.

It is hard to believe that in vivo the Q-modification occurs before splicing. There will be a competition between TGT and TSEN for unspliced tRNA, and tRNA-guanine transglycosylase is known to be a very slow enzyme (k_{cat} rate for murine TGT was reported by Behrens et al., 2018). It might be even worse for a pre-tRNA^{Tyr} substrate (missing data, see below). Therefore, it appears unlikely that Q-modification is faster than splicing.

Major points

1. In order to demonstrate that pre-tRNA^{Tyr} is a substrate for QTRT1/QTRT2, the authors performed an in vitro TGT assay using the ABP gel and LC-MS/MS. However, these data do not show whether the unspliced pre-tRNA^{Tyr} is a reasonably good substrate for QTRT1/QTRT2.

The authors should measure the in vitro enzyme kinetics for pre-tRNA^{Tyr} by using the [8-3H]-guanine incorporation assay. I am really puzzling why this was not done, as the authors did this kind of assay for tRNA^{Tyr}, but not for the pre-tRNA (Fig. S3).

2. The authors performed in vitro binding studies using EMSA and MST. The authors note that these assays were performed in presence of 9dG (page 7, line 176-178).

The presence of 9dG leads to the formation of covalent bond between tRNA and an Asp residue in the catalytic center of QTRT1. Hence, it is not possible to measure a K_d value in the presence of 9dG. There is no equilibrium of association and dissociation as the tRNA stays covalently bound to the protein.

The corresponding paragraph in the Materials and Methods section does not contain any information regarding the addition of 9dG (page 20, line 477 ff.)

Minor points

p. 10, l. 253 What is q30?

Reviewer #3

(Remarks to the Author)

This nice work describes the finding that the microbe-dependent queuosine (Q) tRNA modification in mammals is present in the precursor tRNAs that contain unspliced introns. Q is the only mammalian tRNA modification that depends on a microbe/diet derived metabolite and reports on gut microbiome-host interaction. Although Q-tRNA modification installation, function, and mechanism have been studied for a long time, many questions still remain unresolved. Using cellular, biochemical, and structural biology approaches, this work shows that Q and its downstream galQ modifications are fully present in the unspliced tRNA^{Tyr}. The high resolution structure of the QTRT1/2/pre-tRNA complex offers a molecular rationale for this finding. The manuscript is well written, and the results clearly and logically presented. This study makes a major contribution to revealing and understanding the intricate interplay between tRNA modification and tRNA intron splicing.

Comments:

1. The major unanswered question is whether the Q/galQ-modified pre-tRNA^{Tyr} is an on-pathway product. Does the Q/galQ-modified pre-tRNA^{Tyr} proceed to be spliced, or are these dead-end products that do not get spliced to mature tRNA^{Tyr}? The author's model seems to suggest pre-tRNA to pre-tRNA-Q to tRNA-Q. However, an alternate model could be pre-tRNA to tRNA to tRNA-Q and pre-tRNA to pre-tRNA-Q to degradation. It may be possible to do isotope pulse chase followed by MS measurement to distinguish these models.

2. The results seem to implicate that almost all unspliced tRNA^{Tyr} are present in the cytoplasm. The authors should demonstrate this either by cellular fractionation and/or FISH of the unspliced pre-tRNA^{Tyr}.

Version 1:

Reviewer comments:

Reviewer #1

(Remarks to the Author)

Wei et al. now addressed and answered all points raised, which has improved the manuscript significantly. I have no further comments.

Yadong Sun

Reviewer #2

(Remarks to the Author)

The authors have addressed all critical comments of the reviewer report. Hence, I recommend publication of the revised manuscript.

Reviewer #3

(Remarks to the Author)

The authors have adequately addressed my comments.

Point-by-point response.

Reviewer #1 (Remarks to the Author):

In the manuscript “Queuosine is incorporated into precursor tRNA before splicing”, Guo et al. investigate the temporal order of events in tRNA maturation, focusing on tRNA-splicing and Q-incorporation. The authors observe the presence of Q or galQ -modified pre-tRNAs containing introns, both in vitro and in vivo. The authors also demonstrate the similar recognition mechanisms of mouse QTRT1/2 in complex with mature tRNA^{Tyr} and the intron-containing pre-tRNA^{Tyr} 1-4 by single particle cryo-EM. Overall, this manuscript is well written and well structured. This study provides insights into the highly dynamic and physiological processes involved in tRNA maturation. However, several points need to be addressed before publication in Nature Communications.

We thank the reviewer for the overall positive feedback and the numerous insightful comments, which we address and answer in detail below.

In their study, the pre-tRNA^{Leu} served as a negative control, but interestingly pre-tRNA^{Leu} binds to the mQTRT1/2 complex with similar affinities as pre-tRNA^{Tyr}1-4. To better understand the underlying mechanisms, it would be helpful to provide the AlphaFold 3 prediction of mQTRT1/2 in complexes with pre-tRNA^{Leu}, comparing this model with the complex formed with the intron-containing pre-tRNA^{Tyr} 1-4, to see if there are any differences around active site between the binding modes.

1) Response: Done as requested. We would like to bring to the attention of the reviewer that we used mature m_tRNA^{Leu} as control in the *in vitro* binding assays (revised Figure S5d/e). Murine tRNA^{Leu} indeed binds to mQTRT1/2 complex *in vitro* with similar affinities as observed for tRNA^{Tyr} and pre-tRNA^{Tyr}1-4 (revised Figure 3b), as well as pre-tRNA^{Tyr} 2-1 and m_tRNA^{Tyr} full (revised Figure S5d). Concluding from our results most of these tRNA are potential targets for the QTRT1/2 complex, but the binding of tRNA^{Leu} is observable *in vitro* even though tRNA^{Leu} lacks the modifiable G₃₄ nucleotide. To further investigate the binding of this non-substrate tRNA, we ran AlphaFold 3 prediction for the complex of mQTRT1/2 with tRNA^{Leu} and compared it with the experimentally determined structures of mQTRT1/2 with tRNA^{Tyr} and pre-tRNA^{Tyr} 1-4 (Reviewer Figure 1a). The extended variable loop of m_tRNA^{Leu} does neither clash with the protein nor seem to interfere with the binding – supporting the possibility that this tRNA can bind to QTRT1/2. We hypothesize that under *in vitro* conditions, binding of tRNAs to the protein complex is caused by the highly charged surface, which is responsible for tRNA accommodation. Interestingly, the predicted structural model suggests that recognition by QTRT1 and deformation of ASL of tRNA happens even if the classical GUA motif is missing (Reviewer Figure 1b). Of note, the AF3 models with pre-tRNA^{Tyr} 2-1 (Figure S7), which shows lower activity but also binds with similar affinity to QTRT1/2 (Figure S5d), as well as pre-tRNA^{Leu} 2-1 show an almost identical positioning of the tRNA. As we did not analyse the binding parameters of the later, we prefer to not show the AF3 model with pre-tRNA^{Leu} 2-1 to avoid additional confusions.

We would like to highlight that we have observed the recognition of non-substrate tRNA *in vitro* also for other tRNA modifying enzymes, including the Elongator complex (PMID: 38750017 [1]), PUS3 (PMID: 38996458 [2]), DUS1-4 (unpublished), and the CTU1/2 complex (unpublished). Hence, it seems a more general principle that highly dynamic and processive tRNA modifying enzymes have evolved less stringent selection mechanisms in comparison to other tRNA interacting factors, like amino-acyl transferases. We conclude, that

the complex can in principle bind non-substrate tRNAs and that the selection of tRNAs happens at a later reaction state. This working model is also supported by the observation that different substrate tRNAs (with similar binding affinities) form the covalent intermediate at different rates (new Figure S5a) and show different Q-modification rates (Figure 2) *in vitro*.

Reviewer Figure 1. Predicted AF3 model of mQTRT1/2 tRNA^{Leu} resembles experimentally derived structures despite differences in nucleotide sequence.

a. Comparison of the overall architecture of experimentally derived structures of mQTRT1/2 complexes with pre-tRNA^{Tyr} 1-4 (left), tRNA^{Tyr} (middle) and the predicted structural model of mQTRT1/2 with tRNA^{Leu} (right). Protein domains were colored according to their charge distribution, tRNA is shown in yellow (transparent).

b. Structural comparison (closeup) of the anticodon stem loop of tRNA bound to mQTRT1/2. Surface of the model is shown in white, tRNAs (in analogous order as in panel A) are colored.

The study shows that the presence of intronic sequences does not influence the Q modifications. The mQTRT1/2 complex recognize the exon regions of tRNAs, and the different intron-containing pre-tRNA isoforms bind to the mouse QTRT1/2 complex in similar binding modes (Fig.S6). However, many pre-tRNA^{Tyr} isoforms were expressed but not modified (Table S2). The authors should clarify why certain pre-tRNA^{Tyr} isoforms fail to undergo modification, despite having similar structural features.

2) Response: As pointed out by the reviewer, we have carefully evaluated various aspects of pre-tRNA^{Tyr} expression and its structure, including genomic location and AF3 predictions. We have integrated additional analysis of genomic locations (Table S2) and structural comparisons (see response above) in the revised manuscript. In addition, we would like to highlight that despite similar binding affinities, we do find differences between pre-tRNA^{Tyr} 1-4 and pre-tRNA^{Tyr} 2-1 *in vitro* - please see elaborate response to issue #1 of reviewer 2. In short, the presence of an intron does not affect the overall mode of binding, but different intronic sequences still seems to affect the precise positioning of G₃₄ in the active site to an extent that is different between different pre-tRNAs. Nevertheless, we have not yet performed a systematic analysis for all pre-tRNA^{Tyr} isoforms. We believe that other unknown cell-type specific mechanisms, additional detection methods and selection processes during tRNA maturation might also contribute to the specificity *in vivo*. We fully agree with the reviewer that understanding this aspect on a broader scale is of major importance and we have started

planning systematic follow-up studies to answer these newly discovered aspects of precursor tRNAs, but in our opinion addressing these questions goes beyond the scope of this study.

In vertebrate tRNA^{Tyr}, Q is further glycosylated with galactose and mannose to generate galQ and manQ, respectively. Is it possible that manQ is also added to pre-tRNA^{Tyr} before the splicing of introns ?

3) Response: Done as requested. We tested this hypothesis and no manQ can be detected on mature tRNA^{Tyr} and the tested pre-tRNAs in human and mouse samples. We added these data to the revised Figure S8 in addition to the measurements by mass spectrometry upon β -gal treatment.

Intron-dependent tRNA modification sites are typically modified prior to splicing. The authors show that Q is added to pre-tRNA^{Tyr} before splicing, yet the activity of the mQTRT1/2 complex appears to be higher on mature tRNA^{Tyr} (Fig. 2b). This raises the question of whether Queuosine modification occurs on both intron-containing and intron-lacking substrates,

4) Response: We agree that it is most likely that the modification occurs in both, intron-containing and intron-less substrate tRNA. Hence, the previously included statement (page 14 line 336) in the discussion is still fully valid - “Furthermore, mature tRNA^{Tyr} coming from those precursors, which were never observed to be modified (Table S2), might still undergo modification after being spliced. Our results demonstrate that the presence of intronic sequences does not influence the deformation of the ASL region, imposed upon binding to QTRT1/2 complex.”

and if so, how these two forms of the substrate differ in terms of modification efficiency,

5) Response: Done as requested. We have measured the enzyme kinetics using [8-³H]-guanine incorporation assay for *in vitro* transcribed tRNA^{Tyr}, pre-tRNA^{Tyr} 1-4 and pre-tRNA^{Tyr} 2-1. The results are presented in the revised Figure S4 and are in agreement with the previously presented LC-MS/MS analyses (Figure 2b). As stated in the Results section, page 7 line 162, these results “...show that in *in vitro* conditions, mQTRT1/2 is less active towards pre-tRNA^{Tyr} in comparison to tRNA^{Tyr}” and the value are presented in Figure S4g. In addition, we found similar differences between the isoforms during the formation of the covalent intermediate (revised Figure S4i; see also more elaborate response to issue 1 of reviewer 2).

We would like to highlight that in addition to the *in vitro* transcribed tRNA, we also used bulk tRNA isolated from wild type and Q1 and Q2-knockout mESC in our activity assays. On these endogenous samples, “The enzymatic complex could efficiently modify pre-tRNA^{Tyr} 1-4 as well as mature tRNA^{Tyr} from the pool of unmodified tRNA molecules in Q1 and Q2 mESC (Fig. 2d and e).” Page 7 line 171, which suggest that additional factors can account for efficient modification rate of pre-tRNA^{Tyr} *in vivo*.

and whether the presence or absence of Q modifications influences the splicing process.

6) Response: Done as requested. Similar overall levels of mature tRNA^{Tyr} are found in wild type and Q1 cell lines and tissues of mouse (Figure S2), human (PMID: 30093495 [3]) and other organisms. This fundamental observation suggests that the lack of Q₃₄ would not affect the splicing rates of pre-tRNAs. Nevertheless, we have established an endonuclease activity assay with recombinant human TSEN complex to directly address this question (Figure S3e, e). Indeed, we observed that the presence of Q does not hamper the splicing process, since native pre-tRNA^{Tyr} 1-4 can be efficiently cleaved by TSEN independently of the modification status (see new supplementary Figure S3e). The same applies to human *in vitro* pre-tRNA, shown below in Reviewer Figure 2.

Reviewer Figure 2: TSEN complex activity assay was performed by digestion of both queuosinylated and unmodified *in vitro* transcribed human pre-tRNA^{Tyr} 8-1 with the recombinant TSEN complex as time-course experiment. The input RNA and indicated reaction products were separated by PAGE along with RNA size markers (M).

Minor points:

- Please provide a close-up view of the electron density around the anticodon stem loop (ASL), including the intron region.

7) Response: Done as requested. Closeup views of the density corresponding to intron, compared with analogous region in the structure of mQTRT1/2 with mature tRNA^{Tyr} are presented below in Reviewer Figure 3. Both maps are shown with contour rms-Level 6 and aligned on top of each other. We would like to highlight that we were not able to obtain a clearly interpretable cryo-EM map for the intron and we conclude that the intron remains dynamic. Foremost, the comparative structural analyses of the complex with the mature and the intron-containing tRNA highlights that the intron does not impede with binding to the mQTRT1/2 complex and that it does not interfere with accommodation of the anticodon in the active site. This main conclusion can be clearly drawn from the obtained structure, even if no model of the intronic sequence can be unambiguously built in the obtained map. We would also like to highlight that all cryo-EM data has been deposited to publicly available repositories (EMPIAR, EMDPB and PDB). This will allow interested experts to follow our conclusions and to independently analyse our cryo-EM maps and datasets.

Reviewer Figure 3. Closeup views on the intron density in mQTRT1/2 in complex with tRNA^{Tyr} (left) and with pre-tRNA^{Tyr} 1-4 (right).
 a / b. Side view of the structures with the closeup on intron.
 c / d. Front view of the structures with the closeup on intron.

- Please add a reference to Fig.4a in the main text.

Response: Added as requested.

- Report min/mean/max B factors in Table S1

Response: Done as requested. The respective values have been added to Table S1.

Reviewer #2 (Remarks to the Author):

The hypermodified nucleoside queuosine is introduced in tRNA^{His}, tRNA^{Tyr}, tRNA^{Asp}, and tRNA^{Asn} by the enzyme tRNA-guanine transglycosylase (QTRT1/QTRT2), which replaces the guanine-34 in the anticodon by queuine.

In this manuscript Guo, Kaczmarczyk and colleagues provide evidence that for tRNA^{Tyr} the Q-modification is formed before splicing. They analyzed tRNAs from mouse embryonic stem cells by means of ABP gels and northern blotting. Only a small fraction of below 4% of mature tRNA^{Tyr} contains Q, as it contains galQ, which cannot be detected by the ABP method. Interestingly, 60% of pre-tRNA^{Tyr} shows a signal on the ABP gel, which has been interpreted as Q-content. However, as stated by the authors, the ABP method is not specific for Q, it rather detects cis-diol groups, like in ribose.

The authors performed *in vitro* assays to demonstrate that pre-tRNA^{Tyr} is a substrate for TGT, and they also determined the cryo-EM structure of the pre-tRNA^{Tyr} - TGT complex. This structure closely resembles the tRNA-TGT structure, but the intron is not resolved in the EM reconstruction as it appears to be flexible.

It is hard to believe that *in vivo* the Q-modification occurs before splicing. There will be a competition between TGT and TSEN for unspliced tRNA, and tRNA-guanine transglycosylase is known to be a very slow enzyme (k_{cat} rate for murine TGT was reported by Behrens *et al.*, 2018). It might be even worse for a pre-tRNA^{Tyr} substrate (missing data, see below). Therefore, it appears unlikely that Q-modification is faster than splicing.

We thank the reviewer for the discussion and valuable suggestions, which have contributed to improve and complete the manuscript. Below, we provide detailed responses addressing each point.

Major points

1. In order to demonstrate that pre-tRNA^{Tyr} is a substrate for QTRT1/QTRT2, the authors performed an *in vitro* TGT assay using the ABP gel and LC-MS/MS. However, these data do not show whether the unspliced pre-tRNA^{Tyr} is a reasonably good substrate for QTRT1/QTRT2. The authors should measure the *in vitro* enzyme kinetics for pre-tRNA^{Tyr} by using the [8-³H]-guanine incorporation assay. I am really puzzling why this was not done, as the authors did this kind of assay for tRNA^{Tyr}, but not for the pre-tRNA (Fig. S3).

1) Response: Done as requested. Following the comment, we performed enzyme kinetics using [8-³H]-guanine incorporation assay for *in vitro* transcribed tRNA^{Tyr}, pre-tRNA^{Tyr} 1-4 and pre-tRNA^{Tyr} 2-1. In agreement with the previously presented LC-MS/MS analyses (Fig. 2b), the kinetic measurements show a turnover of mature mouse tRNA^{Tyr} (K_M = 0.23 μM), lower enzymatic turnover for precursor 1-4 and almost no turnover for precursor 2-1 (see Fig S4 b-g). Despite the fact that both analyses show that mQTRT1/2 modifies pre-tRNAs, we were surprised by the reduced kinetics and tried to further analyse the underlying reasons. We have previously shown that the binding parameters for all three tRNAs are very similar in the presence of the inhibitor 9dG. In response to the following comment by reviewer 2 (see below), we measured the binding parameters in the absence of the inhibitor, showing slightly reduced binding affinities for all three tRNAs – apparent K_d of 2.4 μM instead of 1.2 μM for tRNA^{Tyr}; 6.0 μM instead of 2.4 μM for pre-tRNA^{Tyr} 1-4; 2.7 μM instead of 1.4 μM for pre-tRNA^{Tyr} 2-1 (see new Fig. 3b and Fig. S5d). These differences in affinities were expected and do not explain the stark differences in the enzymatic turnover.

Hence, we checked for the formation of the previously described covalent intermediate (PMID: 32149287 [4], PMID: 12949492 [5]) in the presence of the inhibitor. We consider this a measure for accurate accommodation and positioning of G₃₄ in the active site, which in the

presence of 9dG leads to the cleavage of G₃₄ and covalent attachment of the abasic position 34 to a nearby amino acid residue that can be directly detected by denaturing PAGE. Strikingly, the results perfectly align with experimental data from [8-3H]-guanine incorporation assay and LC-MS/MS measurements – showing the formation of a clearly visible intermediate for mature tRNA^{Tyr}, a reduced signal for pre-tRNA^{Tyr} 1-4 and an almost absent signal for pre-tRNA^{Tyr} 2-1. Hence, the presence of the intron does not affect the overall mode of binding, but still seems to affect the precise positioning of G₃₄ in the active site to an extent that is different between different pre-tRNAs. Of note, the detected *in vivo* activity towards pre-tRNA might still be influenced by factors that are not present in our *in vitro* setup (e.g. other tRNA modifications, tRNA binding proteins, post-translational modifications). We have added the data in Figure S4, added the description of the assay to the Methods section and discuss the results in the revised manuscript 7.

2. The authors performed in vitro binding studies using EMSA and MST. The authors note that these assays were performed in presence of 9-deazaguanine (page 7, line 176-178).

The presence of 9dG leads to the formation of covalent bond between tRNA and an Asp residue in the catalytic center of QTRT1. Hence, it is not possible to measure a K_d value in the presence of 9dG. There is no equilibrium of association and dissociation as the tRNA stays covalently bound to the protein.

2) Response: Done as requested. We fully agree with the comment by the reviewer, and we have remeasured all quantitative MST assays without addition of 9dG. In the absence of the inhibitor, we observed similar and only slightly reduced binding affinities for all three tRNAs – apparent K_d of 2.4 μM instead of 1.2 μM for tRNA^{Tyr}; 6.0 μM instead of 2.4 μM for pre-tRNA^{Tyr} 1-4; 2.7 μM instead of 1.4 μM for pre-tRNA^{Tyr} 2-1. We have revised Figure 3, where we now exclusively focus on tRNA^{Tyr} and pre-tRNA^{Tyr} 1-4. We show the MST binding curves for both tRNA in the presence and absence of 9dG and list the apparent K_d values – we believe that the term “apparent K_d” reflects the situation appropriately. As we have used 9dG to increase the binding for the cryo-EM analyses, we would still like to show the EMSA in the presence of 9dG. We have moved the analyses of pre-tRNA^{Tyr} 2-1 to Figure S5, where we now also show the MST data with and without 9dG for tRNA^{Leu} and pre-tRNA^{Tyr} 1-4 with its 5'-leader and 3'-trailer sequences. We have removed the EMSA (Fig. S5c) and the input control (Fig. S5e) in the presence of 9dG to improve clarity. We have updated the Methods section and incorporated the new MST data into the revised manuscript text on page 8.

The corresponding paragraph in the Materials and Methods section does not contain any information regarding the addition of 9dG (page 20, line 477 ff.)

3) Response: Done as requested. We apologise for the confusion and have corrected the appropriate section in the method section.

Minor points

p. 10, l. 253 What is q30?

Response: Clarified as requested. Reference 30 has been moved to avoid confusion

Reviewer #3 (Remarks to the Author):

This nice work describes the finding that the microbe-dependent queuosine (Q) tRNA modification in mammals is present in the precursor tRNAs that contain unspliced introns. Q is the only mammalian tRNA modification that depends on a microbe/diet derived metabolite and reports on gut microbiome-host interaction. Although Q-tRNA modification installation, function, and mechanism have been studied for a long time, many questions still remain unresolved. Using cellular, biochemical, and structural biology approaches, this work shows that Q and its downstream galQ modifications are fully present in the unspliced tRNA^{Tyr}. The high resolution structure of the QTRT1/2/pre-tRNA complex offers a molecular rationale for this finding. The manuscript is well written, and the results clearly and logically presented. This study makes a major contribution to revealing and understanding the intricate interplay between tRNA modification and tRNA intron splicing.

We appreciate the reviewer's positive feedback and have thoroughly addressed their comments in the response below.

Comments:

1. The major unanswered question is whether the Q/galQ-modified pre-tRNA^{Tyr} is an on-pathway product. Does the Q/galQ-modified pre-tRNA^{Tyr} proceed to be spliced, or are these dead-end products that do not get spliced to mature tRNA^{Tyr}? The author's model seems to suggest pre-tRNA to pre-tRNA-Q to tRNA-Q. However, an alternate model could be pre-tRNA to tRNA to tRNA-Q and pre-tRNA to pre-tRNA-Q to degradation. It may be possible to do isotope pulse chase followed by MS measurement to distinguish these models.

1) Response: We have established a TSEN complex activity assay and incorporated the results into the revised manuscript (Figure S3). We have used this assay to demonstrate that the presence of Q does not hamper the splicing process, since native pre-tRNA^{Tyr} 1-4 can be efficiently cut in the presence or absence of the modification (see new Figure S3). We come to the same conclusion for *in vitro* transcribed/modified human pre-tRNA (see also our response to a major point of reviewer 1).

Our *in vivo* observations (page 6 line 141) show “an increase of pre-tRNA fragments in differentiated neurons lacking Q (Fig. S2a). This finding is further corroborated by an increase of pre-tRNA^{Tyr} 1-4 fragments in mouse tissues, including the brain (Figure S2c).” This observation suggests a model where pre-tRNA-Q are stabilized whereas tRNA-Q are more subjected to degradation especially in brain tissues. These results are further discussed on page 14 line 350: “we observed that the absence of Q leads to an increase of intron-containing pre-tRNA^{Tyr} fragments in mouse tissues and neuronally differentiated cells. These aberrant fragments could indicate that Q modification is necessary for the stability of the pre-tRNA (similar to what was observed for mature tRNA and as previously described for cancer cells⁵⁸), or for preventing defects in the splicing mechanism²⁴”. In addition, in our Q1 conditions we never observed an accumulation or an increase of pre-tRNA^{Tyr}, therefore, we consider the model pre-tRNA to pre-tRNA-Q to degradation not supported by *in vivo* observation.

Furthermore, formal proof of the alternate model, in which pre-tRNA to pre-tRNA-Q to tRNA-Q, at the moment appears to be technically difficult, if not impossible, to achieve *in vivo*. The approach suggested by the reviewer, namely to perform an isotope pulse-chase followed by MS measurement to distinguish the two models, seems impractical for addressing our question. MS measurements will not discriminate which RNA harbors Q *in vivo* (pre-tRNA^{Tyr}, mature

tRNA^{Tyr} or other queuinylated tRNAs), and purifying a specific pre-tRNA in a quantity sufficient for mass spectrometry from cells or tissues is not possible at the moment. Nevertheless, to try to answer this interesting and valid question, we attempted a queuine pulse-chase followed by APB Northern or gdAPB Northern blotting (see Reviewers Figure 4, below) and encountered a similar issue to the one described above for the MS approach. While it is possible to detect Q on mature tRNA even after a relatively short pulse (5 minutes treatment with queuine in mESC previously deprived of Q-tRNA), detecting the modification on the lower-expressed pre-tRNAs under the same treatment and shorth pulse proves to be challenging due to their relative abundance and the distinct dynamics these RNA species undergo. Since a substantial portion of mature tRNA^{Tyr} appears to be already modified after just 5 minutes, it is practically impossible to establish a reliable *in-cellulo* experiment capable of distinguishing the fraction of modification originating from pre-tRNA-Q > tRNA-Q from the fraction possibly introduced directly onto the highly abundant mature tRNA.

Reviewers Figure 4. Queuine pulse-chase. mESCs were first depleted of Q-tRNA by cultivating for two weeks with medium without Q or queuine (q) source, using dialyzed FBS. Then the depleted culture was subjected to the indicated queuine (200 nM) pulse and chase inputs.

a Depleted mESCs were subjected to pulse of queuine at the indicated times. The queuinylated m_pre-tRNA^{Tyr} 1-4 and mature tRNA^{Tyr} were detected using APB and gdAPB northern blotting respectively.

b Depleted mESCs were subjected to pulse and chase of queuine at the indicated times. The queuinylated m_pre-tRNA^{Tyr} 1-2 and mature tRNA^{Tyr} were detected using gdAPB northern blotting.

In conclusion, our *in vivo* observations argue against the model in which pre-tRNA transitions to pre-tRNA-Q and then to degradation. Instead, they support both possibilities, which are not mutually exclusive, of pre-tRNA to tRNA and then to tRNA-Q, as well as pre-tRNA to pre-tRNA-Q and then to tRNA-Q. The two possibilities are both mentioned in the manuscript page 14 line 336 and discussed above in a response to Reviewer 1.

2. The results seem to implicate that almost all unspliced tRNA^{Tyr} are present in the cytoplasm. The authors should demonstrate this either by cellular fractionation and/or FISH of the unspliced pre-tRNA^{Tyr}.

2) Response: Done as requested. We have performed cellular fractionation, which revealed the presence of pre-tRNA^{Tyr} 1-4 in the cytoplasm. The data are presented in the new Supplementary Figure 3.

References

1. Abbassi, N.E., et al., *Cryo-EM structures of the human Elongator complex at work*. Nat Commun, 2024. **15**(1): p. 4094.
2. Lin, T.Y., et al., *The molecular basis of tRNA selectivity by human pseudouridine synthase 3*. Mol Cell, 2024. **84**(13): p. 2472-2489 e8.
3. Tuorto, F., et al., *Queuosine-modified tRNAs confer nutritional control of protein translation*. EMBO J, 2018. **37**(18).
4. Alqasem, M.A., et al., *The eukaryotic tRNA-guanine transglycosylase enzyme inserts queuine into tRNA via a sequential bi-bi mechanism*. Chem Commun (Camb), 2020. **56**(27): p. 3915-3918.
5. Xie, W., X. Liu, and R.H. Huang, *Chemical trapping and crystal structure of a catalytic tRNA guanine transglycosylase covalent intermediate*. Nat Struct Biol, 2003. **10**(10): p. 781-8.